# HOW ALIGNED ARE DIFFERENT ALIGNMENT METRICS?

**Jannis Ahlert**[1]* **Thomas Klein**[1,2]* **Felix Wichmann**[1] **Robert Geirhos**[3]

## ABSTRACT

In recent years, various methods and benchmarks have been proposed to empirically evaluate the alignment of artificial neural networks to human neural and behavioral data. But how aligned are different alignment metrics? To answer this question, we analyze visual data from Brain-Score (Schrimpf et al., 2018), including metrics from the model-vs-human toolbox (Geirhos et al., 2021), together with human feature alignment (Linsley et al., 2018; Fel et al., 2022) and human similarity judgements (Muttenthaler et al., 2022). We find that pairwise correlations between neural scores and behavioral scores are quite low and sometimes even negative. For instance, the average correlation between those 80 models on Brain-Score that were fully evaluated on all 69 alignment metrics we considered is only 0.198. Assuming that all of the employed metrics are sound, this implies that alignment with human perception may best be thought of as a multidimensional concept, with different methods measuring fundamentally different aspects. Our results underline the importance of integrative benchmarking, but also raise questions about how to correctly combine and aggregate individual metrics. Aggregating by taking the arithmetic average, as done in Brain-Score, leads to the overall performance currently being dominated by behavior (95.25% explained variance) while the neural predictivity plays a less important role (only 33.33% explained variance). As a first step towards making sure that different alignment metrics all contribute fairly towards an integrative benchmark score, we therefore conclude by comparing three different aggregation options.

## 1 INTRODUCTION

A central question in the field of representational alignment is whether two given perceptual systems apply the same transformation to their inputs, thereby extracting equivalent representations from data. These systems could be artificial deep neural networks (DNNs) and biological systems like primate brains, or arbitrary other image-computable models. For a number of reasons, the perceptual alignment of artificial to biological systems has seen growing interest in recent years. Well-aligned DNNs could serve as models of biological visual processing and ultimately be an important step towards building neural prosthetics. Furthermore, measuring discrepancies between machine and biological perception can help to identify shortcomings in DNNs and improve machine perception (Wichmann & Geirhos, 2023). For example, aligning DNNs with the human visual system promises to increase their robustness and generalization abilities (Dapello et al., 2022; Sucholutsky & Griffiths, 2023).

Since the question of representational alignment between brains and machines cannot be answered theoretically, various empirical methods have been proposed to measure the representational alignment of different systems. These efforts include the comprehensive Brain-Score benchmark (Schrimpf et al., 2018; 2020), which integrates 51 different metrics to capture the alignment of different models to biological systems. In addition to neural data extracted from primates, other metrics also capture behavioral similarity, such as error pattern analysis (Rajalingham et al., 2018; Geirhos et al., 2020; 2021) and shape bias (Geirhos et al., 2018; Baker et al., 2018; Hermann et al., 2020). Other commonly used datasets in the literature include behavioral similarity judgements (e.g., Hebart et al., 2023; Muttenthaler et al., 2022).

In light of this wealth of different metrics and datasets, all of which intend to quantify how "human-like" (or brain-like, or primate-like) machine learning models are, a fundamental question arises:

---

*Equal contribution. [1]University of Tübingen, Germany [2]Max Planck Institute for Intelligent Systems, Tübingen [3]Google DeepMind, Toronto, Canada. Correspondence to: t.klein@uni-tuebingen.de

**How aligned are different alignment metrics?** For instance, if two metrics are highly correlated, it may be sufficient to evaluate models on one of them instead of both. On the other hand, if two metrics lead to highly dissimilar results, they measure different aspects and would lead to very different conclusions and model rankings—assuming the metrics are sound. In contrast, if inconsistencies between different metrics were for instance caused by methodological errors that do not reflect different aspects of brain-likeness, those cases could equally be surfaced by analyzing the agreement of metrics as we do in this study. In order to determine how aligned different alignment metrics are, we therefore analyze a broad set of up to 241 models evaluated on 50 different metrics, depending on data availability.

## 2 METHODS

**Nomenclature.** In the following, we will refer to any function that maps a model to a scalar *score* measuring some form of alignment as a *metric* or *measure*. Any set of metrics that together estimate alignment is referred to as a *benchmark*. The point of this distinction is that for a benchmark, some method of integrating its constituent metrics is necessary, like taking their average.

**Data sources.** A central hub for collecting and integrating similarity measurements between DNNs and biological neural data is the Brain-Score benchmark (Schrimpf et al., 2018; 2020). We use data from Brain-Score as the foundation of our analyses, but also integrate data from three other sources: Muttenthaler et al. (2022), Fel et al. (2022) and Geirhos et al. (2021).

The Brain-Score benchmark (Schrimpf et al., 2018) provides comparative data for two domains, vision and language. Throughout this paper, we focus exclusively on vision and thus on the visual Brain-Score. Brain-Score evaluation results can be thought of as a table, where the columns are different metrics and the rows are models evaluated on these metrics. The columns are semantically grouped into three sets: Neural, behavioral and engineering benchmarks. The neural benchmark can be further subdivided into groups that measure a model's similarity to different brain areas: V1, V2, V4 and IT. The behavioral benchmark consists of image classification metrics like error consistency, either for humans or monkey subjects. The engineering benchmark, which does not enter the final scoring, captures technical properties of models like ImageNet top-1 accuracy.

We are particularly interested in the agreement between neural and behavioral measures of similarity between humans and machines, so we use data from Geirhos et al. (2021), who record human classification performance on (corrupted) ImageNet images and calculate multiple types of behavioral similarity: *error consistency* (Do humans and machines make mistakes on the same images?) and *shape bias* (When presented with shape-texture cue conflicts, do models prefer shape, like humans?). Muttenthaler et al. (2022) systematically analyze which design choices affect the human-machine alignment for various DNNs, finding that neither architecture nor model scale are relevant, but that the training data and objective function are driving factors of alignment. They quantify human-machine alignment as a network's ability to predict human behavior on triplet tasks (Hebart et al., 2020), for which human subjects are presented with three natural images and classify one of them as the odd-one-out. Hence, we call this metric *odd-one-out similarity*. Fel et al. (2022) define human-machine alignment as the similarity between model attention maps and attention maps obtained from human crowd-workers. Linsley et al. (2018) collected about 500k of these human attention maps, which quantify which regions of an image humans attend. We refer to this metric as *attention map similarity*. See Appendix A.2 for a detailed breakdown of models used in our analyses.

**Pairwise comparisons.** To perform pairwise comparisons between two benchmarks, we calculate correlations between their respective scores for the same set of models. We calculate Spearman's rank correlation, which we believe to be appropriate since we are interested in the ranking of models (rather than measuring linear relationships as Pearson's correlation does). We apply very conservative Bonferroni-corrections to the thresholds of significance to account for multiple comparisons. Specifically, we divide the threshold of $\alpha = 0.05$ by the number of metric-pairs, resulting in $\alpha = 0.0009$ for Figure 1 and $\alpha = 0.001$ for Figure 5.

**Integrating metrics: arithmetic mean, z-transformed mean, mean rank.** Given a set of metrics, what is the best way of aggregating them to a single, unified score? While a one-size-fits-all solution is unlikely to exist, we observe that aggregation choices can influence conclusions and thus

warrant attention. Brain-Score uses a simple arithmetic average; in Section 4 we explore alternatives. A typical technique for dealing with different scales involves z-transforming values before aggregating them, which enforces that all individual metrics have zero mean and unit standard deviation. Alternatively, a ranking of models at some level of the Brain-Score hierarchy could be achieved by first ranking all metrics individually, and then calculating the average rank for every model.

## 3 How aligned are different metrics?

**Correlations between Brain-Score benchmarks.** To investigate the internal consistency of metrics on Brain-Score, we calculate Spearman's rank correlation coefficient for every pair of metrics, taking only those 95 models into account for which all scores are available. Removing duplicates and one model with an ImageNet accuracy below 1.9% leaves 88 models. We find that behavioral measures correlate strongly, especially those that all use the same dataset (Geirhos et al., 2021). For many neural metric groups such as V4 and IT, as well as sub-groups within the V1 region relating to texture and receptive field size, internal consistency is higher than consistency with other metrics. We relegate detailed results to the Appendix, see Figure 4.

**Moving beyond Brain-Score.** Next, we investigate how the two new metrics that are not yet included in the Brain-Score project, odd-one-out similarity and attention map similarity, correlate with metrics on Brain-Score in Figure 1. We again calculate pairwise Spearman's rank correlations between metrics for the set of models for which scores are available on all metrics. The odd-one-out similarity metric by Muttenthaler et al. (2022), which is of behavioral nature, is more correlated with metrics of V1 than with other behavioral metrics or later neural areas such as IT. The attention map similarity by Fel et al. (2022) is not significantly correlated with any other metric.

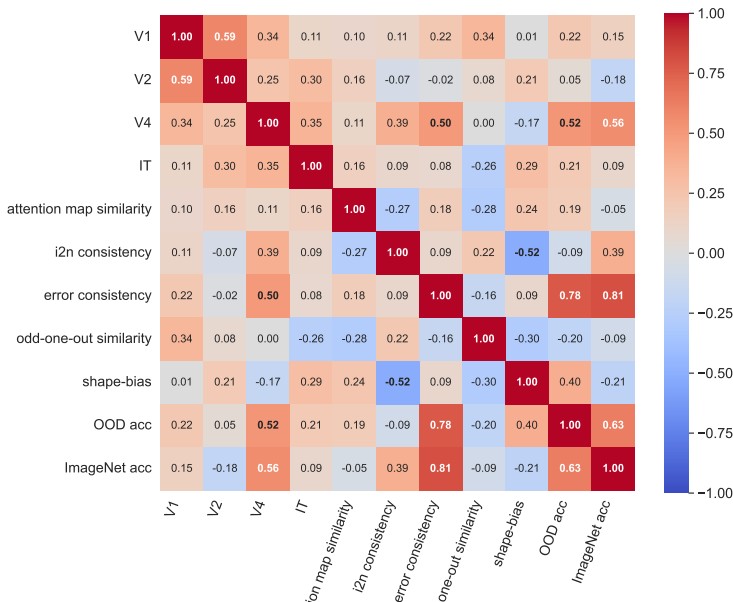

Figure 1: **How aligned are different alignment metrics?** Pairwise Spearman's rank correlations of different Brain-Score metrics, as well as odd-one-out similarity judgements and attention map similarity. Correlations that are significant after Bonferroni-correction are bold. We include only those 42 models that were evaluated on all metrics. Note that (a) variance of correlation coefficients is quite high and (b) similar metrics tend to agree, with the exception of the odd-one-out similarities. See also Figure 5 for the 80 Brain-Score models that have all scores on the Brain-Score metrics of this heatmap.

## 4  ALTERNATIVES TO THE ARITHMETIC AVERAGE:
## HOW DO AGGREGATION CHOICES INFLUENCE ALIGNMENT RESULTS?

The overall score that a model achieves on Brain-Score is calculated as the arithmetic mean of its neural and behavioral scores. In order for this to work well, the two values should at least have roughly the same magnitude, and ideally live in a common metric space. In Figure 2, we scatter behavioral against neural scores, demonstrating the differences in their variance. Note also Linsley et al. (2023), who find that models which perform better on ImageNet (which is a behavioral metric) tend to have worse IT-alignment (a neural metric). It is evident that the best behavioral scores of about 0.6 are quite a bit higher than the best neural scores of 0.5. Because of different scaling of the metrics, models with mediocre neural scores can still rank highly in the benchmark by virtue of their good behavioral scores. In the overall model rank, we find examples of this: The two best models on Brain-Score, a Convolutional Vision Transformer [1] and a ResNeXt variant [2] by Mahajan et al. (2018) are only the 71st and 75th best model according to neural scores, respectively.

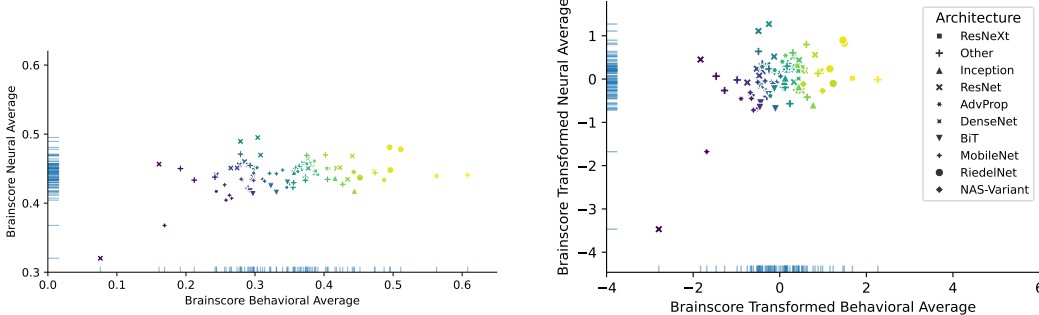

Figure 2: **Left: Relationship between neural and behavioral scores.** Coloring represents the model rank in Brain-Score, which is the average of the neural and behavioral scores (brighter color indicates higher overall score). **Right: Same data after z-transforming both scores.** The dominance of behavioral scores prevails, because there are extreme outliers in the neural scores.

Assuming that we have obtained a matrix of model scores on different, possibly uncorrelated metrics with different distributions, we should ask ourselves how to best integrate these scores into a final model-score, or at least into an overall ranking of models. We now consider three different possible ranking schemes. The first strategy, which is the default in Brain-Score, is to integrate metrics via their arithmetic mean, which is sensitive to scaling issues and high-variance metrics. As an alternative, we consider *z-transforming* the score for every metric before averaging. We demonstrate how this scheme would change scores in Figure 2b. A drawback of this approach is that such scores are no longer stable over time, but depend on the scores of other models evaluated on that metric. Alternatively, if one does not care much about absolute model scores but only about their relative order, a possible integration scheme is obtaining the *rank order* for every metric, to then average over these ranks for the final score. Such a scheme would drop quantitative information about differences, but avoids inconsistency issues like the examples with high overall rank despite extremely low-rank performance on some metrics. In practice, applying these ranking schemes to Brain-Score does indeed lead to changes in model ordering (see Figure 3). Each row corresponds to a ranking scheme, and each model is represented as one point. A model's position on the x-axis corresponds to its score: Good models are further on the right, worse models further on the left, with scores normalized to span the range $[0-1]$. To visualize the impact of the ranking scheme, we color-code models by their position in the default ranking, with higher scores colored brighter. Arguably, the differences in the resulting rank orderings are not "dramatic". However, if further studies were to e.g. correlate brain-likeness against other quantities, the effects of such changes could be quite large. We believe the issue of how to aggregate metrics deserves more scrutiny in the future.

---

[1] https://www.brain-score.org/model/vision/1885
[2] https://www.brain-score.org/model/vision/646

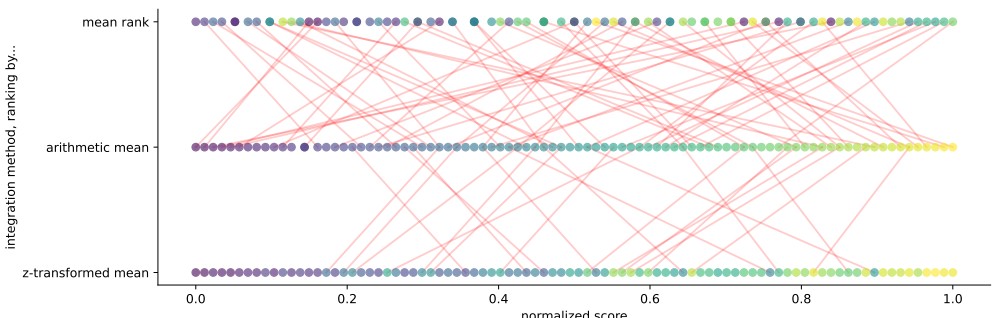

Figure 3: **Comparison of rankings resulting from different integration schemes.** We compare three different ranking algorithms: (1) Arithmetic mean corresponds to the current standard in Brain-Score, where scores are simply averaged. (2) The z-transformed ranking is obtained by z-transforming scores first, then averaging them. (3) Mean Rank is the result of averaging the ranks implied by the metrics (ranks are inverted for consistency, so higher is still better). All scores are normalized to the interval $[0 - 1]$ using Min-Max Normalization and colored according to a model's position under the original integration scheme. Rank-order changes greater than 10 positions (relative to the rank implied by the arithmetic mean) are highlighted in red. Spearman's rank correlations between the original ranking and the new ones are 0.47 (mean rank) and 0.92 (z-transformed mean).

## 5 DISCUSSION

Recent years have seen a wealth of datasets, benchmarks and metrics to measure the alignment between brains and machines. This is a promising research avenue, but in light of an ever-expanding set of measures, this raises the question of how aligned different alignment metrics truly are. We have contributed an analysis of the relationships between different alignment measures. The resulting correlations clearly indicate that different metrics capture different aspects of perceptual alignment, sometimes leading to contradictory results. For instance, models that score as highly human-like on one metric may be among the worst models according to a different metric. Consequently, alignment with human perception may best be thought of as a multidimensional concept, and as a community we may need to spend more time thinking about how to properly integrate different metrics. As a first step in this direction, we have investigated how different aggregation choices influence alignment results. Going beyond simple aggregation methods, in which metrics are treated equally and independently, one could also consider aspects like the semantic structure of metrics: Should we discount a model's high IT-score if it performs poorly on V1 and V2, indicating that it arrives at the right solution via the wrong path? Ultimately, it could be valuable for the community to derive a set of axiomatic requirements that an ideal integration scheme should fulfill, and categorize existing aggregation choices according to how well they conform to those. Overall, it seems clear that while there is a need for unified and comprehensive benchmarks, it may be important to consider alignment as an inherently multidimensional concept.

ACKNOWLEDGMENTS

The authors would like to thank Martin Schrimpf for providing raw Brain-Score data, Alexander Riedel for providing checkpoints of his models, Britta Dorn for inspiring ranking algorithms based on computational social choice, Priyank Jaini and Nina Wiedemann for feedback on the manuscript, as well as Simon Kornblith, Lukas Muttenthaler and Martin Schrimpf for discussions. Furthermore, we would like to thank the anonymous reviewers for their constructive feedback.

Funded by the Deutsche Forschungsgemeinschaft (DFG, German Research Foundation) under Germany's Excellence Strategy—EXC number 2064/1—project number 390727645 and SFB 1233—project number 276693517. This work was supported by the German Federal Ministry of Education and Research (BMBF): Tübingen AI Center, FKZ: 01IS18039A. TK acknowledges financial support via an Emmy Noether Grant funded by the German Research Foundation (DFG) under grant no. BR 6382/1-1, awarded to Wieland Brendel. The authors would like to thank the International Max Planck Research School for Intelligent Systems (IMPRS-IS) for supporting Thomas Klein.

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

# A  APPENDIX

## A.1  LIMITATIONS

Our analyses are limited by several technical issues. First, most of the pairwise correlations calculated in Figure 1 are not statistically significant, because so few models were present in all datasets. There also is some uncertainty about whether model weights were matched perfectly.

## A.2  METHODOLOGICAL DETAILS (MODEL SELECTION)

At the time of writing, Brain-Score consists of $241$ models. However, not all of those models were evaluated on all metrics: Only $88$ different models were evaluated on all metrics that contribute to the final score (i.e. all neural and behavioral metrics). Completing this evaluation would be valuable to allow broader analyses, as would adding exact pointers to model checkpoints where this is feasible.

For the analysis in Figure 1, we determine a subset of the models on Brain-Score for which weights are publicly available and evaluate them on the metrics of Fel et al. (2022) and Muttenthaler et al. (2022). This restriction limits the scope of our analysis and we hope to increase this sample in the future. Thus, the models currently evaluated on all considered metrics are:

1. AlexNet (Krizhevsky et al., 2012) via torchvision
2. DenseNet 121, 169 and 201 (Huang et al., 2017) via torchvision
3. EfficientNet B0 and B7 (Tan & Le, 2019) via torchvision
4. Inception V1 (Szegedy et al., 2015) and V3 (Szegedy et al., 2016) via torchvision
5. Inception V4 (Szegedy et al., 2017) via timm
6. VGG 16 and 19 (Simonyan & Zisserman, 2014) via torchvision
7. ShuffleNetV2-x1.0 (Ma et al., 2018) via torchvision
8. Robust ResNet50-$\ell_2$ with $\epsilon = 1$ and $\epsilon = 3$ from (Salman et al., 2020)
9. 13 MobileNetV2 variants (Sandler et al., 2018)
10. 6 BiT-S Variants by (Kolesnikov et al., 2020)
11. 6 AdvProp-EfficientNet variants by Xie et al. (2020)
12. ResNet50 trained on SIN, SIN & IN, SIN & IN & finetuned on IN from (Geirhos et al., 2018)

## A.3 Further Results

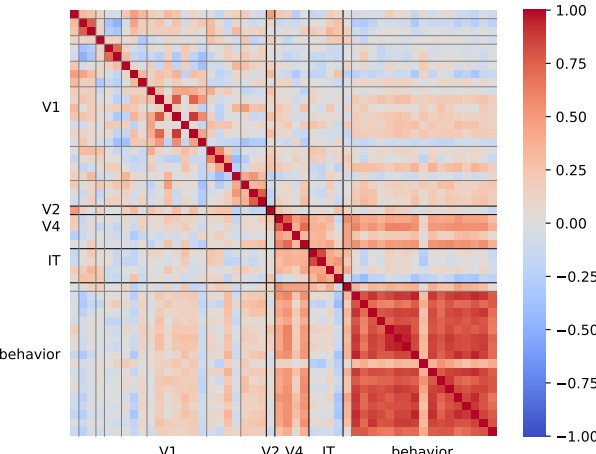

Figure 4: **How consistent are Brain-Score metrics?** Pairwise Spearman's rank correlations of different metrics, from V1 to IT and finally to behavioral measures. Note how the agreement between the behavioral metrics is much higher than the agreement of the neural measures.

To calculate the correlations presented in Figure 1, we included all models that were evaluated both on Brain-Score and the metrics proposed by Muttenthaler et al. (2022) and Fel et al. (2022). However, this set of models is relatively small (42 models total), so the correlations reported between the different Brain-Score metrics are not necessarily representative. We demonstrate how Figure 1 would have looked like if all Brain-Score models had been included in Figure 5.

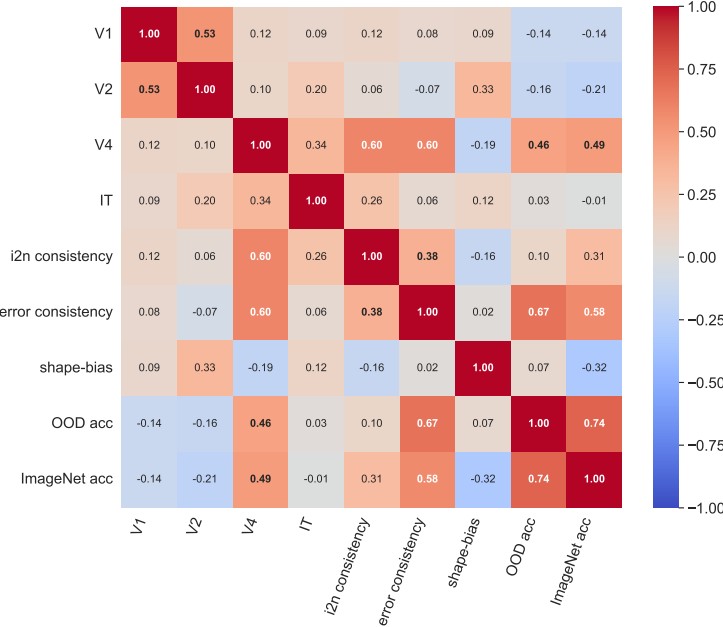

Figure 5: **Pairwise Spearman's rank correlations of Brain-Score metrics, from V1 to IT and behavioral measures.** In contrast to Figure 1, we include all Brain-Score models that had a value available for each metric, not only the subset that was also evaluated on the metrics by Muttenthaler et al. (2022) and Fel et al. (2022). This amounts to a total of 80 models. The average correlation on this heatmap is 0.17, while the average correlation of the smaller set of models in 1 is 0.21, exemplifying the need for further thorough evaluations.

The neural score that a model obtains on Brain-Score is defined as the average over four metrics measuring alignment to different brain regions (V1, V2, V4, IT). This averaging strategy will work well if those scores are highly correlated and similarly distributed. We therefore investigate the distributions of neural scores in Figure 6, revealing similar distributions after removal of models with incomplete evaluations.

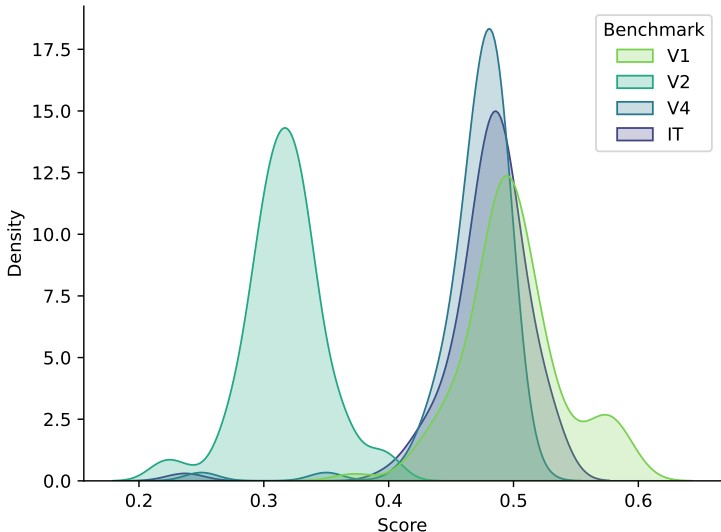

Figure 6: **Kernel Density Estimates of different neural metrics in Brain-Score.** This plot includes those 88 models for which all scores that contribute to the overall Brain-Score ranking are available. Evidently, the distribution of scores for areas V1, V4 and IT are very similar, but the scores for area V2 are much lower, either hinting at lack of progress towards predicting V2 or at unfair calibration of this metric.

While we find that behavior already accounts for a disproportionate amount of the overall score variance on the current Brain-Score leaderboard, we next investigate which role missing values play in determining the aggregate scores Figure 7. Coloring each model by the number of non-zero scores available across metrics reveals a striking pattern: Most models fall into one of a few homogeneous groups, having 50, 33, 30 or 5 non-zero metrics respectively. Presumably, not all authors re-triggered scoring when new chunks of metrics were added to Brain-Score. The plot reveals that behavioral scores account for even larger portions of variance when only comparing models that have the same number of non-zero metrics with each other.

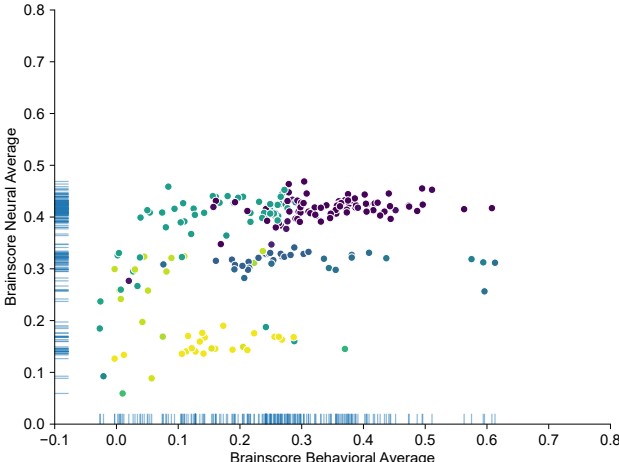

Figure 7: **How do missing scores shape Brain-Score?** Scatterplot of neural and behavioral averages for all models on Brain-Score. Models are colored according to the number of benchmarks they have been evaluated on, as determined by counting scores that are not exactly zero. Darker colors indicate more non-zero scores, yellow models have only been evaluated on 5 of 51 possible metrics.

