# OpenReview forum: "How aligned are different alignment metrics?"
_ICLR.cc/2024/Workshop/Re-Align — ICLR 2024 Workshop Re-Align Poster_

### Official Review · Reviewer_yfAF · 2024-02-12
**A thought-provoking topic, shy of solid conclusions, but with high community interest.**

**Rating:** 2
**Fit:** 3
**Confidence:** 3

**Review:**

[field to be removed]

**Workshop Review:**

The authors tackle a very interesting question, how aligned are multiple different metrics by looking at a large set of models on Brain Score’s 51 metrics. Metrics are universally acknowledged as imperfect and while it’s known that different metrics are sensitive to different features of the data (and transformations of said data) it’s entirely understandable that metrics do not align along the same dimensions and can give conflicting results for the same models. The authors suggest that by studying aggregation techniques, a fairer representation might give more insight into common aggregation metrics in place today. The authors are also open to the idea that some metrics might be flawed and do not automatically deserve a place in a hypothesised perfect ranking aggregation.

The last paragraph of Section 2 is a bit unclear to me. I’m not sure what ‘hierarchy’ is being referred to and this means the logic of applying the mean to all such models doesn’t make much sense to me. However, I assume this point will be clarified later in the paper (it wasn’t). I can see results were described as “not significant” without reference to alpha level, exactly what is being included in the Bonferroni correction etc. I can’t tell if the threshold was 0.01 and the tests reported 0.03, which would have passed a 0.05 threshold. It’s not even possible to get a vague sense of if this is even a limited data issue because we’re not told by how much the the predictions are away from being significant (without getting into the Fisher / Neyman-Pearson debate on p-values here).

Some more general thoughts:

Why should either “neural” or “behavioural” be weighted equally (playing Devil’s Advocate)? Should we not care about one over the other? These feel like questions we should ask ourselves in the community and I like the fact this paper brings up the issue for consideration. However, are the proposed methods (z-scoring / ranking) the best ones? Without a clear definition for what we want as a community out of leader boards like Brain-Score, we’re just changing one ranking for another without being sure it’s inherently better because we don’t have external guidance to qualitatively say that we should care equally (or not) about average scores on neural or behavioural tasks.

My take on the paper is that it brings up an important issue that I believe is worthy of consideration and could promote good discussion within the community when speaking of metrics generally. My reservations are that it principally points out a problem with the Brain-Score platform (specifically: arithmetic mean of (i) neural and (ii) behavioural scores). This is a limited domain that doesn’t necessarily apply across the field generally and is up to the creators to decide how they would like to rank / score candidate models. Beyond that, the proposed changes of z-transforming the scores / ranking introduce issues which the authors clearly and openly outline in the paper itself. This leaves the question still open as to whether such a shift in the final ranking is necessarily "better" or "more desirable". I can envisage scenarios where rank ordering is undesirable and the absolute values (i.e. not-ranked) scores are better metrics to work with. There’s still the case where we treat neural metrics as separate to behavioural ones and do away with unified metrics for the reasons claimed in the paper.

Having said that, I certainly feel thankful to the authors for bringing this issue to the surface and forcing me to think through what I would expect in alignment metrics and highlighting the potential for misinterpretation of models that perform highly on Brain-Score, since such models could have uneven performance across multiple domains. I also wasn’t aware of such a high correlation between the “behavioural” metrics and wonder now what they’re all measuring at a more fundamental level. Will this lead to statements like, “Our model scores well on X% of the metrics on Brain-Score?" where X is an inflated percentage, and hide the fact there is a large confound measuring broadly the same effect in the data? It would have been great if this were a long paper that went a bit more into the micro (and less macro) details to explore some of these questions.

I think the paper could be greatly enhanced even with a few modifications. Particularly, I'd like to hear some arguments about the downstream effects of what the effect would be of making the suggested changes in many scenarios that we might run into, in the community. For example, if someone needed to pick a model for a concrete downstream scenario, and they load up Brain-Score benchmark and have a look, what aspects about what they care about would be the most affected by each of your proposed metric changes? I also think you might want to look at page 5 of the Brain-Score paper on bioarxiv, which mentions an effect of taking the arithmetic mean across multiple metrics and why ranking might not be the best metric. In Figure 6, the same issue is sort of hinted at, but I originally thought it was a criticism of how Brain-Score had implemented things, but upon further checking, this is something discussed in the original paper with some (small) rationale.

Why was this not submitted as a long paper? The main bulk of your results are all in the appendix and reorganisation could have brought these into the main paper where you could have had better organisational capacity to arrange things in a better format. Right now there is a "Further Results" appendix section that flows as a whole section's worth of text and figures. I think Figure 7 is your paper's main "message" but it's been bundled in the appendix while I think it deserves more prominence in the main paper. It's also not clear why you have decided to mark ranking changes (in red) that have moved more than 10 slots, without quantification of why this number makes sense. What does this tell us? This isn't comparable if there were 10% as many models, so maybe shifted position out of the total number of models might make a bit more sense here.

Clarity:

Very well-written and pleasant to read. Useful definitions provided (the "nomenclature" section) and only one or two very small places where I tripped up and wasn't sure of the intended meaning. Overall, very clear. Appendix section feels a little less refined, and also not sure why it's there as there was scope for a longer paper submission that could have had those figures and texts in there to still be under that page limit where things are better organised by sections and supporting text for the figures.

Correctness:

This is a little harder to evaluate specifically, because an issue was pointed out and the proposed 'fixes' are not really aligned with any external criteria and no clear judgement on a recommendation was given in the end. It had more of a thought-provoking essence to it than analysing a correct conclusion. Some missing statistical information in the paper, which I have never seen omitted before. However, this is in terms of showing lack of significance and is less serious than claiming significance without revealing that information.

Novelty:

I certainly was made aware of some issues I hadn't fully considered and for me I see novelty. However, I can imagine others who are very familiar with Brain-Score who might not see as much novelty in it. The methods proposed (z-transform, ranking) are not necessarily new or interesting of themselves, but the application to how we rank candidate models of brain / behaviour is formulated in light of one of our sole large-scale benchmarks, so overall I would say yes, this work does exhibit relevant novelty.

Interest to the community:

Driven mainly by my response to "Novelty" and being a member of the community, I can say I'd be interested to see this work if I wasn't currently reviewing it. How broadly the community will find it interesting is not a point I can answer, but I think it will promote discussion and is definitely targeted at the community of people looking at representational alignment.

**Reason For Not Giving Higher Score:**

I gave 2 for Workshop Rating (poster) and the reason I don't think it fits best as a talk is because of the lack of solid conclusions and conveying a key piece of targeted advice we can all take away from the work. It's best for people, if they are interested in measuring their models with Brain-Score, to take the time to visit that poster specifically.

**Reason For Not Giving Lower Score:**

It's clear this piece of work was aimed at exactly this community and contained interesting things to say. That means I couldn't possibly be unsure whether this work fits the community (it does) and I think there is enough merit to the work to have it above the acceptance threshold (particularly in light of reviewer guidelines urging to interpret evaluation sections widely & generously).

**Reviewer Domain:**

cognitive science

---

> ### Author Response · Authors · 2024-05-03
> **Thank you for your review!**
>
> Thank you for writing such a thorough and constructive review! We especially appreciated your feedback on reporting correlations and significance strengths and have updated the respective plots to better reflect which results are significant. We have also evaluated more models on these metrics, so that more values have become significant. Regarding our methodology and statistical details, that were indeed missing before: We obtain the list of model scores for each metric and then, for every pair of metrics, calculate the pairwise Spearman’s correlation coefficient between their two lists. We apply a very conservative Bonferroni-correction to these pairwise correlations, dividing the significance level by the number of pairs. We have updated the figures to represent which specific correlations were significant.
>
> _“Why should either neural or behavioral scores be weighted equally?”_ This is a great question, and we hope that our paper can spark an open debate about how different types of alignment scores should be weighted when integrating them, or whether we should integrate scores in the first place.
>
> _“The last paragraph of Section 2 is a bit unclear to me. I’m not sure what ‘hierarchy’ is being referred to.”_  We apologize for the confusion - we were referring to the hierarchy of scores in Brain-Score, where scores are grouped hierarchically (e.g. the neural-score consists of the V1, V2, V4 and IT scores, each of which in turn aggregates other scores). We have updated the manuscript to improve clarity.
>
> _“...points out a problem with the Brain-Score platform”_ This is technically correct, since we really draw most of our data from Brain-Score, because it is by far the biggest and most popular benchmark of DNN-brain-alignment. But we also include two data sources that are not (yet) present on Brain-Score, and we believe that the points we raise about integrative benchmarking generalize beyond Brain-Score.
>
> _“...arguments about the downstream effect of making the suggested changes”_ Thank you for this great point. The downstream effects of different ranking algorithms would be the changed ordering of models, which could result in different selections of models for other experiments. Also, correlations of “brain-likeness” against other quantities would be massively different. We have added this concrete example to the manuscript.
>
> _“I can envisage scenarios where rank ordering is undesirable and the absolute values are better”_ This is a fair point - depending on the concrete research question, the scores themselves are sufficient. Rank-ordering models only becomes relevant in the context of benchmarking, as done on Brain-Score.
>
> _“Why was this not submitted as a long paper?”_ In retrospect, that might have been a good idea - we are very happy to hear that you found this work interesting enough to warrant a longer treatment.

---

### Official Review · Reviewer_tGuJ · 2024-02-27
**Important work on the alignment of different human-machine alignment metrics.**

**Rating:** 2
**Fit:** 3
**Confidence:** 3

**Workshop Review:**

Overall I found the study to be very well-motivated, tackling an important issue (heterogeneity between metrics measuring human-machine alignment), and the work has clearly documented that the rank orders of models across metrics is not consistent.

I had only a few minor issues/questions/comments to note:
1) the choice of spearman correlation makes sense for rank ordering models, but it seems to me that getting the magnitudes aligned is equally important. If one metric shows very high alignment for the best models (say .8) and the other very low alignment for the best models (say .2), even if they have the same rank orders of models, I would say these metrics aren't in agreement. You might consider Lin's Concordance, a measure of agreement that's sensitive to both the covariance and absolute agreement between measures (and varies between -1 and 1).

2) It seems import to compute and report noise-corrected scores. Take for instance the correlation between v4 and IT (.3), and v4 versus i2n consistency (.51). Is the v4 metric in greater alignment with the i2n consistency metric than with the IT metric? Numerically yes, but the difference could be due entirely to differences in measurement error / reliability. For instance, if both brain-metrics had a noise ceiling of .3, then the maximum observable correlation between them would be .3. If the i2n consistency metric is more reliability measured (behavioral data can be more reliable if aggregating over many subjects, so reliability .85 isn’t implausible), then it would have a maximum observable correlation of about .51. In other words, it’s possible v4 agrees with IT and i2n consistency equally well and that the apparent difference in those agreement strengths is due to differences in measurement noise.

\[r_{true} = \frac{r_{observed}}{\sqrt{reliability\_a \times reliability\_b}}\]

So the noise ceiling can be estimated as sqrt(reliability_a x reliability_b)

It would be helpful to see, side-by-side with the current heat maps, the heat-maps with noise-adjusted correlations (r_observed / noise_ceiling).

3) The text doesn’t make it very clear that the model attention maps were used to predict brain activations (I’m assuming that’s the case).

4) It would be helpful if the authors could offer suggestions on how these metric-alignment/misalignment scores can be understood beyond the main point (degree of alignment is low). For example, it seems that model agreement increased for many of the metrics for the analysis in Figure 2 versus Figure 1. What are we to make of those increases in agreement for the attention-map based analyses relative to the feature-based ones?

5) Unfortunately the study seems to be under powered, as noted in the Figure captions many of the reported correlations (even high ones) are not statistically significant. What statistical procedure was used? Some visual indication of which correlation scores are significant and which are not would be helpful.

**Reason For Not Giving Higher Score:**

I think this work is on the right track, but the current study is underpowered (many of the reported correlations are not significant, including the highest ones, e.g., between behavioral measures). There's also a bit more that can be done on the analysis front to nail down the magnitude of disagreement taking noise-ceilings into account, and more importantly on the conceptual front to clarify what the take away point of differences in agreement should be. These issues are related, since some differences in agreement between metrics must only reflect differences in measurement noise for the different metrics. But even with noise taken into account, what would we make of certain metrics being in higher agreement with each other than others? I wouldn't expect the authors to get to the bottom of this in this one paper, but it would be helpful for the authors to discuss the issues that we need to contend with (e.g., some metrics use flexible linear-regression methods, others don't, etc.)

**Reason For Not Giving Lower Score:**

Although more work needs to be done to nail this down, the work that was reported seems like a solid first step.

**Reviewer Domain:**

cognitive science

---

> ### Author Response · Authors · 2024-05-03
> **Thank you for your review!**
>
> Thank you for your highly constructive review. We especially appreciate your insightful comments on the statistics, such as incorporating Lin’s Concordance and reporting noise-corrected scores.
>
> _“The current study is underpowered”_ We were indeed limited by the number of models that were evaluated on both Brain-Score and the respective new metrics at the time of submission. In the meantime, we have evaluated more models on those metrics ourselves, and have learned that a large set of models will be added to Brain-Score soon (unfortunately not in time for the workshop). We will further update our manuscript once more data is available, but the added models have already allowed us to obtain significant results (see new figure 1).
>
> _“You might consider Lin's Concordance”_ Thank you for this interesting proposal. We have looked into Lin’s Concordance and might re-do some of our plots in future iterations of the manuscript, but for now will stick with Spearman’s rank correlation, as it’s more commonly known and more intuitively clear, and because it is robust against different noise ceilings (see next question).
>
> _“It seems important to compute and report noise-corrected scores.”_ You are absolutely correct that noise ceilings should be taken into account when comparing metrics. The Brain-Score platform is aware of this need and reports corrected scores for most of its metrics, and the Attention-Map similarity is also corrected for this. Some other metrics, e.g. behavioral Brain-Score scores and zero-shot similarity by Muttenthaler et al. are not corrected, but since we calculate Spearman’s Rank Correlation, our heatmaps should be inherently robust against this issue.
>
> _“The text doesn’t make it very clear that the model attention maps were used to predict brain activations (I’m assuming that’s the case).”_ We apologize for the confusion. The model attention maps from Fel et al were not used to predict brain activations. Instead, they were compared to human attention maps (collected from crowd workers) to judge similarity between humans and DNNs. We have updated the manuscript to better explain this method.
>
> _“What statistical procedure was used? Some visual indication of which correlation scores are significant and which are not would be helpful.”_ Thank you for this very sensible suggestion. We have updated our figures to reflect which correlation scores are significant and clarified the wording regarding our methodology. In a nutshell: We obtain the list of model scores for each metric and then, for every pair of metrics, calculate the pairwise Spearman’s correlation coefficient between their two lists. We apply a very conservative Bonferroni-correction to these pairwise correlations, dividing the significance level by the number of pairs.
>
> _“It would be helpful if the authors could offer suggestions on how these metric-alignment/misalignment scores can be understood … What are we to make of those increases in agreement for the attention-map based analyses relative to the feature-based ones?”_ Thank you for this question. In former figures 1 and 2, we were considering two metrics that are not yet included in Brain-Score, but also evaluate alignment between DNNs and brains in some way. The differences between figures 1 and 2 appear because different subsets of the models on Brain-Score were used to calculate the alignment between Brain-Score and the data by Muttenthaler et al and Fel et al. We have since been able to evaluate more of the Brain-Score models on the other two metrics, so that we can now plot this comparison in a joined figure (see new figure 1).

---

### Author Response · Authors · 2024-05-03

We would like to thank all reviewers for taking the time to review our paper so thoroughly and providing such constructive and helpful feedback. We are delighted to read that reviewers found the study to be “well motivated, tackling an important issue” (tGuJ) and consider our research question “very interesting” (yfAF).
We have updated our manuscript to incorporate as many of the suggested improvements as possible, especially with regards to how we report significance. The camera-ready version of the paper includes more models evaluated on the different benchmarks to increase the statistical power of our analyses. We also evaluated enough models on the two new metrics to be able to merge former figures 1 and 2 into a common heatmap.
Additionally, we updated former figure 3 to only include those models that were actually evaluated on all benchmarks, which both seems fairer and happens to illustrate our point better.

---

### Decision · Program_Chairs · 2024-03-02

Accept (Poster)